# Clinical-Epidemiological Profile of Dental Professionals Associated with COVID-19 Infection in Southern Peru: A Cross-Sectional Study

**DOI:** 10.3390/ijerph20010672

**Published:** 2022-12-30

**Authors:** Caroline Suarez-Cabello, Erick Valdivia, Andrea Vergara-Buenaventura

**Affiliations:** 1Facultad de Ciencias de la Salud, Universidad Cientifica del Sur, Panamericana Sur Km 19, Villa, Lima 15067, Peru; 2Department of Periodontology, School of Dentistry, Universidade Federal do Rio Grande do Sul, Porto Alegre 91501-970, Brazil; 3Sección de Periodoncia e Implantes, Departamento de Estomatología, Hospital Central de la Fuerza Aérea del Perú, Miraflores 15046, Peru

**Keywords:** COVID-19, dentistry, infection, occupational health, oral health, prevalence, SARS-CoV-2

## Abstract

Dental professionals have been identified as being at high risk for COVID-19 infection due to close contact with patients and the nature of dental treatments. However, the prevalence of infected dentists in Peru has not been determined. An online electronic survey was sent to dentists registered with the College of Dentists of Arequipa to collect sociodemographic data, medical conditions, and employment characteristics during the COVID-19 pandemic. The clinical characteristics and adverse effects of dentists diagnosed with COVID-19 were also recorded. The overall prevalence of COVID-19 infection was 44%. The highest number of diagnosed patients ranged in age from 31 to 40 years (*n* = 111; 48.9%). A total of 45.9% of female and 41.6% of male dentists were diagnosed with COVID-19 (*p* = 0.425). A relationship was found between the district of origin and infection, and dentists working in the public sector during 2021 showed a greater trend of becoming infected (57.6%) (*p* < 0.05). The antigen swab test was the diagnostic test most frequently used (38%), and cough was the symptom most reported. Only 0.6% reported being hospitalized, 10.6% reported needing oxygen during hospitalization, and 0.6% were admitted to an intensive care unit. As in previous studies in other countries, the rate of COVID-19 infection among Peruvian dentists is high. It is recommended that dentists carry out infection control measures while ministries of health and dental associations take measures to ensure their safety.

## 1. Introduction

During the coronavirus disease (COVID-19) pandemic, many healthcare workers (HCW) acquired the disease while working with infected individuals [1,2]. In fact, it has been reported that HCW have a higher risk of infection than other essential occupations and other individuals in the community [3]. The main route of person-to-person transmission has shown to be via airborne droplets from an infected person or a contaminated surface [4], with direct or indirect transmission by many asymptomatic people spreading the virus in environments in which aerosols are generated [5]. Because the oral cavity is considered a reservoir for severe acute respiratory syndrome coronavirus 2 (SARS-CoV-2) and saliva a source of contamination, dentists were no exception to acquiring and transmitting the disease due to the close contact with patients and the nature of dental treatments [6,7]. On the other hand, the incubation period of the infection has shown that people are most infectious in the first week post SARS-CoV-2 infection, when showing few or no symptoms, thereby complicating its control [8].

Dental clinics may have been a high-risk environment for disease spread due to the frequent production of aerosols and the presence of saliva during dental procedures [7]. The rates of COVID-19 reported among dentists identified dental practices as being at high risk [9]. The nature of dental treatment, coupled with the inapplicability of social distancing and the care of unmasked patients, pose a high-risk scenario [10]. On the other hand, the COVID-19 pandemic uncovered several deficiencies in the dental care system, especially related to a lack of personal protective equipment (PPE) and insufficient coordination of health services [11].

On 6 March 2020, Peru reported the first confirmed case of COVID-19 and became one of the countries most affected by the epidemic [12]. The subsequent rise in cases reflected the lack of minimum requirements for infection control and guidelines for the prevention and management of the disease [13]. Moreover, poor access to health services in terms of policies, infrastructure, and distribution of human resources had a high impact on health services [14]. All these deficiencies led to Peru becoming one of the countries with the highest number of cases and deaths due to the pandemic [15]. In view of these risks, dentists faced several dilemmas regarding the risks/benefits of continuing dental care during the pandemic [16].

Few studies have reported the incidence of SARS-CoV-2 infection among dentists [17,18], nor have the clinical and epidemiological characteristics in Latin American countries been reported. There is also a lack of clarity about the prevalence of SARS-CoV-2 infection among dental professionals based on specific clinical settings, thereby limiting the possibility of planning preventive measures to reduce virus transmission. Therefore, the present study aimed to describe the prevalence, clinical characteristics, and adverse outcomes of SARS-CoV-2 infection among Peruvian dentists. The findings of this work could help plan for better prevention in future pandemic events.

## 2. Materials and Methods

### 2.1. Study Design

An observational, descriptive, cross-sectional study was conducted between 1 January and 31 March 2022. The primary objective of the survey was to determine the prevalence, clinical-epidemiological characteristics, and adverse outcomes of SARS-CoV-2 infection among dentists in the region of Arequipa in southern Peru using an online survey.

### 2.2. Study Population and Sample Size Calculation

The population consisted of dentists registered with the Arequipa College of Dentists a professional association that registers and accredits dentists in that region. In order to be allowed to work, a licensed dentist must be registered with the institution.

The sample size was calculated with a 95% confidence level and a 5% error. Considering that around 3200 dentists were registered, the minimal sample size calculated was 344, with an expected acceptance prevalence of 50% for the variables associated with the outcome. The sample used was limited to dentists registered with the Arequipa College of Dentists, residing in the region between 2020 and 2021 during the COVID pandemic, and who had completed the survey. Sampling was by convenience and using a snowball sampling method.

### 2.3. Survey

An electronic online Google Forms^®^ survey was sent to all the dentists registered with the Arequipa College of Dentists. The institution sent the survey through the e-mails registered in its database. Sociodemographic data (age, gender, and provenance district), chronic medical conditions, and allergies were collected. Employment characteristics during 2020 and 2021 (laboral sector, type of job, and dental specialty) were also collected. Finally, the clinical characteristics (signs and symptoms) and adverse effects reported by dentists diagnosed with COVID-19 were recorded.

The assigned survey required approximately 5 min to complete. All questions were mandatory, which ensured the absence of incomplete answers. In addition, participants were told not to answer the survey if they were not dentists and not to respond to the survey twice, thereby reducing the risk of duplicate responses. Records that were double filled were deleted from the database. The survey was disseminated through the College of Dentists of Arequipa on 9 March 2022. After 20 days, a reminder was sent via institutional e-mail. The survey was closed on 20 May 2022.

### 2.4. Ethical Considerations

This study was conducted in accordance with the Declaration of Helsinki and was registered under protocol 445-2021-POS99, and it was subsequently evaluated and approved by the Institutional Ethics Committee of the Universidad Cientifica del Sur. Before starting the survey, participants were required to sign an electronic informed consent form informing them that their participation was voluntary and unpaid and giving information about the potential risks and benefits. In addition, they were assured that all data extracted from the survey would remain confidential and anonymous.

### 2.5. Statistical Analysis

Continuous variables between the diagnosed and undiagnosed COVID-19 groups were described as mean ± standard deviation. Categorical variables were described as numbers (%), and proportions were compared using the Chi-square test.

All the hypothesis tests were two-sided, and a *p*-value < 0.05 was determined as the level to reject the null hypothesis. The analyses were performed with the SPSS version 25. 

## 3. Results

A total of 408 valid surveys were collected, corresponding to approximately 12% of the dentists in Arequipa (*n* = 3200). Of these, 179 dentists had been diagnosed with COVID-19 and 229 had not, with the overall prevalence of COVID-19 infection among dentist being the 44%.

### 3.1. Sociodemographic Characteristics of Surveyed Dentists

The mean age of the dentists diagnosed with COVID-19 was 40 (SD = 13.4) years, which was not significant compared to that of the 106 uninfected dentists (41.5 ± 5 years). The age group most frequently diagnosed with COVID-19 ranged from 31 to 40 years of age (48.9%; *n* = 111). There was a higher frequency of female dentists surveyed (*n* = 218), of whom 45.9% (*n* = 100) indicated having been diagnosed with COVID-19 at some point during the pandemic. However, no significant differences in sex were found for COVID-19 infection. On the other hand, there was a relationship between the district of origin and the development of the disease; 64.7% (*n* = 11) of dentists working in Paucarpata and 61.4% (*n* = 35) in Cayma were more likely to be infected, and those working in La Joya (22.2%) and Sachaca (33.3%) were less likely to be infected (*p* = 0.041) (Table 1).

The distribution of the presence of comorbid diseases or allergies did not significantly differ between infected and noninfected dentists (*p* > 0.05), with the majority of dentists diagnosed (*n* = 159) or not diagnosed (*n* = 198) with COVID-19 reporting the absence of medical conditions. The most common comorbidity in diagnosed dentists was obesity (60%), followed by arterial hypertension (33.3%) (Table 1).

### 3.2. Job Characteristics of Dentists in 2020

A high number of respondents (*n* = 357) answered that they had done some dental care during the pandemic in 2020. The frequency of dentists diagnosed with COVID-19 was highest among those working in emergency care and part-time, with 46.7% and 49.1%, respectively (*p* = 0.028). A very low percentage of dentists (6.9%) reported not working during 2020. No relationship was found between the risk of infection according to employment in the public or private sector (*p* ≥ 0.05) (Table 2).

### 3.3. Job Characteristics of Dentists in 2021

On the contrary, in 2021, a relationship was found regarding the employment sector. Dentists who worked in the public sector showed a greater trend of becoming infected by COVID-19 (57.6%) (*p* < 0.05); however, there were no differences in relation to the type of dental job (*p* ≥ 0.05). A total of 3.18% of dentists reported not working during 2021. Regarding dental specialties, orthodontists (*n* = 63) and endodontists (*n* = 63) more frequently worked during the pandemic. However, those involved in oral implantology were more likely to be infected by SARS-CoV-2 (Table 2).

### 3.4. Clinical Characteristics and Adverse Effects of Dentist Infected by SARS-CoV-2

The diagnostic test most used for the diagnosis of SARS-CoV-2 infection was the antigen swab test (38%), followed by polymerase chain reaction (PCR) (27.4%) and the presentation of symptoms (8.9%). In comparison, only 1.7% were diagnosed by saliva testing. Among the respondents, 41.3% (*n* = 74) reported having been diagnosed between July and December 2020. Of those who were infected, 15.6% (*n* = 28) reported having no signs, and 13.4% (*n* = 24) reported no symptoms. On the other hand, cough and a combination of shortness of breath, fever, and cough were the signs most reported, with 17.9% (*n* = 32) and 14.5% (*n* = 26), respectively. On the other hand, fatigue and pain were the most reported symptom (23.5%). Less common symptoms were feeling tired and alteration of senses (4.5%) (Table 3).

When evaluating the adverse outcomes of COVID-19 infection, only 0.6% (*n* = 1) reported having been hospitalized, while 10.6% (*n* = 19) reported needing oxygen and 0.6% (*n* = 1) were admitted to an intensive care unit (Table 3).

Regarding reinfection, 90.5% (*n* = 162) were infected only once, 8.9% (*n* = 16) twice, and 0.6% (*n* = 1) more than twice. Regarding vaccination, 15.6% of infected dentists reported having been vaccinated with two doses and 83.8% with three doses by 2022.

## 4. Discussion

The present study aimed to describe the prevalence, clinical characteristics, and adverse effects of SARS-CoV-2 infection reported by Peruvian dentists. The overall prevalence of the disease among dentists was 44%. Similar studies in other countries have shown heterogeneous results, with the prevalence of infected dentists in Europe being between 1.9% [18] and 25.4% [19,20], the prevalence in Brazil being 4.9% [10], and the lower rates of 0.9% and 1.1% being reported in the United States [21] and Italy, respectively [22]. These differences in rates may be due to the periods during the pandemic during which the surveys were carried out. In addition, each country implemented different public health actions in response to the pandemic to prevent the transmission of SARS-CoV-2 [10]. Moreover, these results are only as accurate as the COVID-19 diagnostic tests, which are subject to false-positive and -negative results [23].

The highest number of diagnosed patients was found in the age range of 31 to 40 years. As in other studies, the dentists who were infected belonged to the 30–55 age range [22,24]. We found no significant difference in gender in relation to the incidence of COVID-19, which is similar to the results of other studies [25,26]. However, most of the literature has reported an association between male gender and higher infection rates [27,28,29].

We found an interesting relationship between the district of origin and the frequency of infection. People with lower socioeconomic levels (income, employment, and education) present a lower health status than those with a higher socioeconomic status [30]. Living in disadvantaged areas, working in front-line or high-exposure professions, living in large and numerous households, having comorbidities, and having poor access to health services have been linked to playing a key role in how communities are affected [31].

Although in this study, the distribution of the presence of comorbid diseases or allergies did not significantly differ between infected and noninfected dentists, the most frequent comorbidity was obesity, followed by arterial hypertension, which is similar to the results of previous studies [18,21]. However, the mean age of the study cohort was 36 years, and the young age of the individuals might underestimate the influence of comorbidities on SARS-CoV-2 infection.

A high frequency of dentists carried out some dental care in 2020. During the pandemic, mental health was primarily related to financial issues and fear of infection. Many dental practices had temporary closures, leading them to seek financial assistance during the pandemic [32]. Previous studies have reported that the main challenges faced by dentists during the COVID-19 pandemic were lower patient volume, fear of contracting the disease while working, and feeling less prepared to work [10,16,33]. In addition, studies in Latin America indicated that professionals who worked only in private clinics considered that there was a significant reduction in their monthly income [34].

The highest frequency of dentists infected with COVID-19 was found in those working in emergency care or part-time. This could be explained by the fact that dentists working as front-line employees as well as those who had the need to work failed to achieve social distancing during the pandemic to reduce SARS-CoV-2 transmission [30]. One of the main approaches adopted to prevent SARS-CoV-2 transmission among HCW and the local population was the use of PPE involving mainly respiratory equipment such as face masks and face shields [35]. In this sense, SARS-CoV-2 infection has been associated with poor compliance with infection control guidelines and a lack of PPE [36]. On the other hand, in 2021, a relationship was found with the employment sector. Dentists who worked in the public sector showed a greater trend of becoming infected with COVID-19. Unfortunately, not all hospitals were able to provide adequate PPE to their staff, especially early in the pandemic [37].

In addition, dentists who worked in the public health care system in countries such as Brazil were reported to have a greater fear of contamination [34]. However, these results do not coincide with those of Bachmann et al. [38], who found that dentists in private practices reported a significantly higher rate of infection compared with those working in clinics.

Our study shows that a very low percentage of the dentists surveyed reported not working during 2020 and 2021, with only 3% reporting not having worked in dentistry in 2021. Previous studies have described that intentions to leave work were associated with the risk of exposure to COVID-19, the impact of pandemic management, and access to PPE [39].

One of the causes attributed to the transmission of SARS-CoV-2 in dental settings was exposure to aerosols and droplets, which are generated in oral and maxillofacial surgery [4,6,7]. Our study found that dentists involved in oral implantology were more likely to be infected by SARS-CoV-2. The greatest risk of infection was observed in those who worked in emergency care or part-time, probably because the protocols followed by professionals accustomed to working on a constant basis compared to others only working in emergencies may induce protection errors and lack of strict compliance with PPE measures [35,40].

Similar to other studies, the most commonly used form of testing was the antigen test (38%), followed by PCR (27.4%). Antigen tests are cheaper and faster but may yield false-positive/negative results that can influence the diagnosis [10]. It is important to question the antigen test, as it is known to have poor results in terms of sensitivity. Reverse transcription PCR remains the most accurate form of testing currently available [38,41].

Our results are consistent with those of previous studies, which show that patients infected with SARS-CoV-2 mainly presented with cough, a combination of shortness of breath, fever, and symptoms such as anxiety, fatigue, and pain [25,26,42]. Despite the high number of hospitalizations by COVID-19 and SARS-CoV-2 reported [26], it appears that dentists in Arequipa had a lower frequency, with only one dentist reporting having been hospitalized. While the possible reasons for these manifestations are not yet clear, it could be suggested that certain host genetic factors, as well as viral genetic variations, may influence the clinical course of COVID-19 [43,44].

Regarding reinfection, it was reported that 90.5% were infected only once, 8.9% were infected twice, and 0.6% were infected more than twice, indicating a low rate of reinfection among dentists in Arequipa. It has been suggested that the risk of reinfection may depend on immune status, the severity of infection, cross-immunity, age, and other immunological factors related to T and B cell memory [45]. One of the positive findings of this study was the high rate of vaccination against COVID-19. In Peru, health professionals were prioritized for vaccination in the early stages of the Peruvian immunization program. we observed that 15.6% of infected dentists reported having been vaccinated with two doses and 83.8% with three doses by 2022.

### Study Strengths and Limitations

This study has some limitations related to the use of retrospective surveys, which may induce some degree of quality problems and underreporting. Therefore, the results should be interpreted with caution considering the retrospective nature of the study. On the other hand, in relation to the data of diagnosed patients, all the techniques for SARS-CoV-2 diagnosis (antigen swab, PCR, serological, and saliva test) were included since diagnostic tests in Peru were expensive. However, these tests may yield false-positive/negative results that could influence the diagnosis and the results of the study [10]. Information on the smoking habits of the surveyed dentists was also not available. On the other hand, it should be taken into account that the study design did not allow the determination of cause–effect relationships, and thus, these results should be taken with caution. It should be noted that there is a likelihood that COVID-19 transmission may have occurred in non-dental settings among staff and family members, but due to the recording methodology, this was not delineated.

Despite these limitations, our study has some strengths, including a large sample size that could partly compensate for these limitations. The snowball sampling technique made it possible to increase the sample size and reduce the cost and time of the investigation [46]. In addition, we consider that the results obtained are consistent and close to those observed in other Peruvian health groups during the pandemic.

The purpose of this work was to contribute to global knowledge about COVID-19. It is recommended that dental professionals carry out infection-control measures, taking into account that aerosols and droplets are considered the main routes of the spread of respiratory diseases such as SARS-CoV-2 [47]. Dental professionals should be familiar with the development, spread, and evolution of COVID-19 and other diseases to be able to identify infected patients and take the necessary protective measures [47]. Ministries of health and dental professional associations should take measures to ensure access to PPE [48].

## 5. Conclusions

This study demonstrates the importance of knowing the prevalence, clinical characteristics, and adverse effects of SARS-CoV-2 infection among dentists to develop better public health strategies. With COVID-19 being an as yet unclarified disease, there is a clear need to understand and analyze these data. Rapid identification of at-risk personnel may help reduce transmission in the community and plan for better prevention in future pandemics.

## Figures and Tables

**Table 1 ijerph-20-00672-t001:** Sociodemographic characteristics of the dentists surveyed.

	Prevalence				Total	
	Noninfected		Infected			
	N°	%	N°	%	N°	%
Age						
21 to 30 years old	48	60.0	32	40.0	80	100.0
31 to 40 years old	117	51.1	111	48.9	228	100.0
41 years old and over	64	64.0	36	36.0	100	100.0
*p*	0.048 (*p* < 0.05) *					
Gender						
Female	118	54.1	100	45.9	218	100.0
Male	111	58.4	79	41.6	190	100.0
*p*	0.425 (*p* ≥ 0.05)					
District						
Arequipa	37	56.1	29	43.9	66	100.0
J.L. Bustamante y R.	33	52.4	30	47.6	63	100.0
Yanahuara	14	53.8	12	46.2	26	100.0
Cayma	22	38.6	35	61.4	57	100.0
Cerro Colorado	29	60.4	19	39.6	48	100.0
Hunter	7	53.8	6	46.2	13	100.0
Miraflores	17	63.0	10	37.0	27	100.0
Socabaya	9	56.3	7	43.8	16	100.0
Mariano Melgar	15	75.0	5	25.0	20	100.0
Paucarpata	6	35.3	11	64.7	17	100.0
Alto Selva Alegre	6	60.0	4	40.0	10	100.0
Sachaca	4	66.7	2	33.3	6	100.0
La Joya	7	77.8	2	22.2	9	100.0
Characato	2	50.0	2	50.0	4	100.0
Others	21	80.8	5	19.2	26	100.0
*p*	0.041 (*p* < 0.05) *					
Comorbidities						
None	198	55.5	159	44.5	357	100.0
Arterial hypertension	4	66.7	2	33.3	6	100.0
Obesity	2	40.0	3	60.0	5	100.0
Other	25	62.5	15	37.5	40	100.0
	0.677 (*p* ≥ 0.05)					
Allergies						
No	186	56.5	143	43.5	329	100.0
Yes	43	54.4	36	45.6	79	100.0
*p*	0.801 (*p* ≥ 0.05)					
Total	229	56.0	179	44.0	408	100.0

(*) significance.

**Table 2 ijerph-20-00672-t002:** Employment characteristics of dentists in 2020 and 2021.

2020	Prevalence				Total Noninfected	
	Noninfected		Infected			
	N°	%	N°	%	N°	%
Laboral Sector						
Public	14	46.7	16	53.3	30	100.0
Private	180	58.4	128	41.6	308	100.0
Both	35	50.0	35	50.0	70	100.0
*p*	0.243 (*p* ≥ 0.05)					
Type of Job						
Full-time	23	74.2	8	25.8	31	100.0
Part-time	82	50.9	79	49.1	161	100.0
Emergency care only	88	53.3	77	46.7	165	100.0
Worked in another area	17	73.9	6	26.1	23	100.0
Did not work	19	67.9	9	32.1	28	100.0
*p*	0.028 (*p* < 0.05) *					
2021	Prevalence				Total Noninfected	
	Noninfected		Infected			
	N°	%	N°	%	N°	%
Laboral Sector						
Public	28	42.4	38	57.6	66	100.0
Private	166	59.9	111	40.1	277	100.0
Both	35	53.8	30	46.2	65	100.0
*p*	0.033 (*p* < 0.05) *					
Type of Job						
Full-time	95	52.2	87	47.8	182	100.0
Part-time	111	59.7	75	40.3	186	100.0
Emergency care only	8	57.1	6	42.9	14	100.0
Worked in another area	8	61.5	5	38.5	13	100.0
Did not work	7	53.8	6	46.2	13	100.0
*p*	0.684 (*p* ≥ 0.05)					
Total	229	56.0	179	44.0	408	100.0
Dental Specialty	N°	%	N°	%	N°	%
Does not have	84	57.1	63	42.9	147	100.0
Orthodontics	33	52.4	30	47.6	63	100.0
Periodontics	13	65.0	7	35.0	20	100.0
Oral Rehabilitation	15	55.6	12	44.4	27	100.0
Oral Implantology	6	40.0	9	60.0	15	100.0
Pediatric Dentistry	16	55.2	13	44.8	29	100.0
Endodontics	36	57.1	27	42.9	63	100.0
Cariology	11	61.1	7	38.9	18	100.0
Restorative Aesthetics	12	54.5	10	45.5	22	100.0
Maxillofacial Surgery	3	75.0	1	25.0	4	100.0
*p*	0.943 (*p* ≥ 0.05)					

(*) significance.

**Table 3 ijerph-20-00672-t003:** Clinical characteristics and adverse effects of SARS-CoV-2 in infected dentists.

Characteristics	N°	%
Diagnostic Test		
Antigen swab	68	38.0
Molecular swabbing/PCR	49	27.4
Rapid serological test (digital puncture)	43	24.0
Saliva test	3	1.7
Symptoms	16	8.9
Date of Infection		
March–June 2020	25	14.0
July–December 2020	74	41.3
January–June 2021	31	17.3
July–December 2021	19	10.6
During 2022	30	16.8
Signs		
Did not present	28	15.6
Cough	32	17.9
Difficulty breathing	15	8.4
Fever	13	7.3
Rhinitis	4	2.2
Rhinitis & Cough	16	8.9
Fever & Cough	21	11.7
Shortness of Breath & Cough	17	9.5
Difficulty Breathing & Fever & Cough	26	14.5
Other	7	3.9
Symptoms		
No symptoms	24	13.4
Presence of pain (head. throat. muscle)	34	19.0
Tiredness	23	12.8
Alteration of senses	8	4.5
Tiredness and presence of pain	42	23.5
Presence of pain and alteration of senses	20	11.2
Tiredness and alteration of senses	8	4.5
Fatigue. presence of pain and alteration of senses	20	11.2
Hospitalization		
No	178	99.4
Yes	1	0.6
Oxygen		
No	160	89.4
Yes	19	10.6
ICU		
No	178	99.4
Yes	1	0.6
Number of Infections		
Once	162	90.5
Twice	16	8.9
More than twice	1	0.6

Legend: PCR: polymerase chain reaction; ICU: intensive care unit.

## Data Availability

The data presented in this study are available on request from the corresponding author due to respondent privacy restrictions.

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
