# Peer review of "Clinical-Epidemiological Profile of Dental Professionals Associated with COVID-19 Infection in Southern Peru: A Cross-Sectional Study"

_ijerph, 2022, doi:10.3390/ijerph20010672_

Round 1
Reviewer 1 Report
Dear authors,
The scientific study rises a few concerns:
-the novelty of the topic
-the sample size of the study is very limited- only 12% of the dentists has responded
-there is no evidence that the dentists had the COVID-19 disease from the dental office. How can you link the infection with the dental treatments? From my point of view is impossible.
-the protective measures against the infection may have a positive or negative influence?
-it is possible, due to the heterogeneity of the tests used, that some results to be false positive or can interfere with the study results?
Author Response
Prof. Dr. Paul B. Tchounwou
Editor-in-Chief IJERPH 
Thank you for giving us the opportunity to submit a revised draft of the manuscript entitled "Clinical-epidemiological profile of dental professionals associated with COVID-19 infection in Southern Peru: a cross-sectional study" IJERPH-2098504
We want to thank the reviewers for their time and effort in providing comments which have improved the manuscript. Please find below the point-by-point responses to the reviewers' comments and concerns. All changes in the manuscript have been made with the "track change" tool.
REVIEWER 1
Reviewer #1: Dear authors, The scientific study rises a few concerns:
- the novelty of the topic.
- Response: Thank you for your comment. While we agree that the topic is not novel, COVID-19 research remains a novelty in South American countries, as there are still high rates of infection in the population. The recent pandemic has proved to be a highly contagious disease that has prompted the dental profession to make significant changes in patient care worldwide and the literature has focused on the interrelationship between front-line physicians and COVID-19.
A previous literature search did not find many similar articles on the continent, much less in Peru, and thus, we felt the need to describe the characteristics (clinical and epidemiological) of dentists infected by COVID-19. We believe that having more data on the pandemic will be of great help in possible similar future events.
- the sample size of the study is very limited- only 12% of the dentists has responded
- Response: Thank you for your comment. We agree that the sample size was small, however, it is essential to emphasize that our database included all dentists registered with the Arequipa College of Dentists. This population includes all registered dentists and, therefore, also comprises retired dentists. Nonetheless, one of the inclusion criteria was that the dentists must have worked during the pandemic; thus, we consider that the final sample of 408 valid questionnaires is not limited.
- there is no evidence that the dentists had the COVID-19 disease from the dental office. How can you link the infection with the dental treatments? From my point of view is impossible.
- Response: Thank you for your comment. One of the limitations of COVID-19 research was determining the exact site or route of infection. Further research in dentistry has proposed the oral cavity as the source of infection and aerosol procedures as a risk of transmission, even with unmasked patients who may be asymptomatic. However, determining whether the route of transmission occurred exclusively in the dental office is complex, so we can only speak of a risk of exposure during dental procedures that is amplified due to the open and invasive nature of dental services. Although our study did not aim to determine the source of contagion, it does aim to make dentists aware and adopt the necessary measures to prevent contamination. We have added a paragraph discussing this in Limitations (lines 319 - 333).
- the protective measures against the infection may have a positive or negative influence?
- Response: Thank you for your comment. Protective measures against infection have a positive influence as they are intended to prevent the spread of disease by taking the necessary precautions in dental practice. This topic was discussed in the Introduction (lines 55- 60) and in the Discussion (lines 254-258).
- it is possible, due to the heterogeneity of the tests used, that some results to be false positive or can interfere with the study results?
- Response: Thank you for your comment. Indeed, this is possible and this was discussed and the paragraph has been lengthened in Section 4.1, Study Strengths and Limitations (lines 321 - 327).
Reviewer 2 Report
I would like to greatly thank the authors for their work and effort for their study "Clinical-epidemiological profile of dental professionals associated with COVID-19 infection in Southern Peru: a cross-sectional study". I would like to note down some issues. I would like to see a limitation paragraph. There are several grammar mistakes and the authors should apply the journal guidelines about the references in the text and the punctuation marks.
Author Response
Prof. Dr. Paul B. Tchounwou
Editor-in-Chief IJERPH 
Thank you for giving us the opportunity to submit a revised draft of the manuscript entitled "Clinical-epidemiological profile of dental professionals associated with COVID-19 infection in Southern Peru: a cross-sectional study" IJERPH-2098504
We want to thank the reviewers for their time and effort in providing comments which have improved the manuscript. Please find below the point-by-point responses to the reviewers' comments and concerns. All changes in the manuscript have been made with the "track change" tool.
REVIEWER 2
Reviewer #2: I would like to greatly thank the authors for their work and effort for their study "Clinical-epidemiological profile of dental professionals associated with COVID-19 infection in Southern Peru: a cross-sectional study". I would like to note down some issues.
- I would like to see a limitation paragraph.
- Response: Thank you for your comment. A paragraph on the limitations of the study can be found at the end of the Discussion in Section 4.1, Study Strengths and Limitations.
- There are several grammar mistakes
- Response: Thank you for your comment. The manuscript has again been revised by a native English-speaking language editor.
- the authors should apply the journal guidelines about the references in the text and the punctuation marks.
- Response: Thank you for your comment. The references of the manuscript have been elaborated and corrected following the guidelines of the journal: "In the text, reference numbers should be placed in square brackets [ ], and placed before the punctuation;"
Reviewer 3 Report
Thank you very much for the opportunity to review the work.
The work is very interesting and essential for the development of science. It has a very good structure and presented conclusions.
The following comments are intended to help the authors to make the work even better and more transparent, especially for the international reader.
The work submitted for review addresses an important topic. It is so important that despite the many works that deal with the situation of HCWs during the pandemic, there still needs to be more work that focuses on the situation of dentists during the COVID-19 pandemic. This is important insofar as the contact between dentists and patients during treatment is very close, and the risk of exposure to infection is exceptionally high, requiring certain precautions.
What I write about above is relevant to the paper under review, as I missed drawing attention to it in the Introduction. While it may seem obvious to the authors, I suggest expanding some of the threads with a few detailed sentences, such as those in lines: 39-45.
The part that needs to be improved and supplemented in Chapter 2. on methodology.
Section 2.1 Study Design lacks a definition of how the study was designed. There is more information on this subject in the Abstract than in this section. This section should answer the questions: what was the idea for the study? What was the reason for this group? What characterized it? For what reason in this part of the country? I don't understand why this research was "observational" and cross-sectional if implemented within one city. This section should explain what the College of Dentists of Arequipa is. And how the researchers are connected to it. Table 1 mentions districts. For what reason is this not described in the Study Design?
2.3 Survey I suggest changing the Questionnaire. It also seems that entering Google's data is unnecessary. Was the survey link sent to all dentists? Are the address (email) details of the dentists publicly known? If not, where did the authors of the survey get this data? Lines: 82-84 - was the survey implemented during this period? What do these records mean? Earlier, the authors wrote that the study was implemented in 2021.
There is a massive mess in section 2.1. - 2.3. which does not allow the reader to figure out: when, how the surveys were implemented, by whom, and using which contact list. To how many people was the Questionnaire sent? This undeniably needs to be completed, modified, and cleaned up.
3.1 Patient or Respondents/Denis?
Line 123. In scientific papers of this type, it does not seem reasonable to emphasize the lack of statistical significance; statistical significance is emphasized.
Line 127. the notation of p-value should be uniform throughout the paper; it is usually denoted as p or p
Table 1 Sociodemographic Characteristics of whom? Tables and graphs must be well described - this also applies to other tables. There is no subject in the titles.
3.2. this subsection is repeated
3.2. what do "work characteristics" mean? I would look for another name for these subsections
Line 150. write SARS-CoV-2.
Line 270 I suggest separating this section for discussion with subsection 4.1 Study Strengths and Limitations.
Author Response
Prof. Dr. Paul B. Tchounwou
Editor-in-Chief IJERPH 
Thank you for giving us the opportunity to submit a revised draft of the manuscript entitled "Clinical-epidemiological profile of dental professionals associated with COVID-19 infection in Southern Peru: a cross-sectional study" IJERPH-2098504
We want to thank the reviewers for their time and effort in providing comments which have improved the manuscript. Please find below the point-by-point responses to the reviewers' comments and concerns. All changes in the manuscript have been made with the "track change" tool.
REVIEWER 3
Reviewer #3: Thank you very much for the opportunity to review the work. The work is very interesting and essential for the development of science. It has a very good structure and presented conclusions.
The following comments are intended to help the authors to make the work even better and more transparent, especially for the international reader.
- The work submitted for review addresses an important topic. It is so important that despite the many works that deal with the situation of HCWs during the pandemic, there still needs to be more work that focuses on the situation of dentists during the COVID-19 pandemic. This is important insofar as the contact between dentists and patients during treatment is very close, and the risk of exposure to infection is exceptionally high, requiring certain precautions.
What I write about above is relevant to the paper under review, as I missed drawing attention to it in the introduction. While it may seem obvious to the authors, I suggest expanding some of the threads with a few detailed sentences, such as those in lines: 39-45.
- Response: Thank you for your comment. We agree with you and have added more information on the topic in the Introduction (lines 37 and 45).
- The part that needs to be improved and supplemented in Chapter 2. on methodology.
Section 2.1 Study Design lacks a definition of how the study was designed. There is more information on this subject in the Abstract than in this section. This section should answer the questions: what was the idea for the study? What was the reason for this group? What characterized it? For what reason in this part of the country? I don't understand why this research was "observational" and cross-sectional if implemented within one city. This section should explain what the College of Dentists of Arequipa is. And how the researchers are connected to it. Table 1 mentions districts. For what reason is this not described in the Study Design?
- Response: Thank you for your comment. We have added more information about the study design. We recognize an error in mentioning the city instead of the region. Also, we have added information about the College of Dentists of Arequipa in lines 90-93
- 2.3 Survey I suggest changing the Questionnaire. It also seems that entering Google's data is unnecessary. Was the survey link sent to all dentists? Are the address (e-mail) details of the dentists publicly known? If not, where did the authors of the survey get this data? Lines: 82-84 - was the survey implemented during this period? What do these records mean? Earlier, the authors wrote that the study was implemented in 2021.
- Response: Thank you for your comment. We have removed google data and added information to the section as per your suggestion. The Arequipa College of Dentists sent the survey through the e-mails registered in its database. The survey was conducted in 2021, but work information was recorded for 2020 and 2021. We have edited the information to make it easier to understand.
- There is a massive mess in section 2.1. - 2.3. which does not allow the reader to figure out: when, how the surveys were implemented, by whom, and using which contact list. To how many people was the Questionnaire sent? This undeniably needs to be completed, modified, and cleaned up.
- Response: Thank you for your comment. We have edited this information to make it easier to understand.
- 1 Patient or Respondents/Denis?
- Response: Thank you for your comment. We apologize for the error; it has been corrected.
- Line 123. In scientific papers of this type, it does not seem reasonable to emphasize the lack of statistical significance; statistical significance is emphasized.
- Response: Thank you for your comment. We have edited the information in lines 149-150
- Line 127. the notation of p-value should be uniform throughout the paper; it is usually denoted as p or p
- Response: Thank you for your comment. We apologize for the error; this has been corrected in line 22. The P value has been standardized using a capital letter (P=)
- Table 1 Sociodemographic Characteristics of whom? Tables and graphs must be well described - this also applies to other tables. There is no subject in the titles.
- Response: Thank you for your comment. We have corrected the title of Tables 1, 2, and 3.
- 2. this subsection is repeated
- Response: Thank you for your comment. We apologize for the error and it has been corrected.
- 2. what do "work characteristics" mean? I would look for another name for these subsections
- Response: Thank you for your comment. We have corrected the subsection titles.
- Line 150. write SARS-CoV-2.
- Response: Thank you for your comment. This has been corrected.
- Line 270 I suggest separating this section for discussion with subsection 4.1 Study Strengths and Limitations.
- Response: Thank you for your comment. At your suggestion, we have added this subsection.
Round 2
Reviewer 1 Report
I appreciate the changes that the authors have made and i consider that the manuscript can be published.
Best regards!
Reviewer 3 Report
No suggestions